

# Enhancing infectious disease prediction model selection with multi-objective optimization: an empirical study

Deren Xu[1], Weng Howe Chan[2] and Habibollah Haron[1]

[1] Faculty of Computing, Universiti Teknologi Malaysia, Faculty of Computing, Johor, Johor Bahru, Malaysia
[2] Universiti Teknologi Malaysia, UTM Big Data Centre, Ibnu Sina Institute For Scientific and Industrial Resarch, Universiti Teknologi Malaysia, Johor, Johor Bahru, Malaysia

## ABSTRACT

As the pandemic continues to pose challenges to global public health, developing effective predictive models has become an urgent research topic. This study aims to explore the application of multi-objective optimization methods in selecting infectious disease prediction models and evaluate their impact on improving prediction accuracy, generalizability, and computational efficiency. In this study, the NSGA-II algorithm was used to compare models selected by multi-objective optimization with those selected by traditional single-objective optimization. The results indicate that decision tree (DT) and extreme gradient boosting regressor (XGBoost) models selected through multi-objective optimization methods outperform those selected by other methods in terms of accuracy, generalizability, and computational efficiency. Compared to the ridge regression model selected through single-objective optimization methods, the decision tree (DT) and XGBoost models demonstrate significantly lower root mean square error (RMSE) on real datasets. This finding highlights the potential advantages of multi-objective optimization in balancing multiple evaluation metrics. However, this study's limitations suggest future research directions, including algorithm improvements, expanded evaluation metrics, and the use of more diverse datasets. The conclusions of this study emphasize the theoretical and practical significance of multi-objective optimization methods in public health decision support systems, indicating their wide-ranging potential applications in selecting predictive models.

# INTRODUCTION

Over the past few decades, with the acceleration of globalization and the increase in population mobility, the outbreak and spread of infectious diseases have become a major challenge for global public health (*Fialho et al., 2023*; *Hernández-Giottonini et al., 2023*; *Mirzania, Shakibazadeh & Ashoorkhani, 2022*). From the Severe Acute Respiratory Syndrome (SARS) outbreak in 2003, to the H1N1 influenza pandemic in 2009, and the recent COVID-19 pandemic, outbreaks of infectious diseases have not only had a tremendous impact on human health but have also posed unprecedented challenges to the

Corresponding authors
Deren Xu, 2008xuderen@gmail.com
Weng Howe Chan, cwenghowe@utm.my

world economy and social stability (*Cao et al., 2022*; *Gao, Shang & Jing, 2022*; *Li et al., 2023*; *Yang et al., 2022*). Therefore, effective infectious disease prediction models are crucial for the prevention and control of epidemic outbreaks. They enable public health decision-makers to take proactive measures and mitigate the negative impacts of the epidemic (*Dixon et al., 2022*; *Lv et al., 2021*; *Zhao et al., 2022*).

However, despite significant advances in this field in recent years, existing infectious disease prediction models still have some non-negligible limitations (*Hu et al., 2023*; *Li et al., 2020*; *Liao et al., 2022*; *Tian et al., 2021*). Among them, most models have adopted single-objective optimization approaches, focusing primarily on enhancing prediction accuracy while neglecting other critical factors such as the model's generalizability, computational efficiency, and feasibility in practical applications (*Akbulut et al., 2023*; *Khoo et al., 2024*; *Tsai, Baldwin & Gopaluni, 2021*; *Xia et al., 2022*; *Ye, Li & Zhang, 2020*). This pursuit of a single objective may lead to limitations in the application of the model under specific circumstances, failing to fully meet the complex demands of the public health sector (*Akbulut et al., 2023*; *Sassano et al., 2022*).

In response to these challenges, multi-objective optimization (MOO) offers a new solution. Multi-objective optimization is a method designed to simultaneously optimize multiple conflicting objectives, capable of generating a set of optimal solutions that achieve the best trade-off among the objectives (*i.e.*, Pareto optimal solutions) (*Le Fouest & Mulleners, 2024*; *Mohammed et al., 2023*). In the context of infectious disease prediction, incorporating multi-objective optimization allows models to simultaneously consider prediction accuracy, computational efficiency, and generalizability to new data, thereby enhancing the overall performance of the models (*Feng & Zhang, 2023*; *Liu et al., 2024*). Additionally, multi-objective optimization has been proven to be an effective method for improving decision quality in other fields, such as engineering design, resource allocation, and environmental management (*Huang et al., 2024*; *Wang, Zhao & Zhang, 2023*).

This study aims to explore and empirically demonstrate the application of multi-objective optimization methods in selecting infectious disease prediction models, addressing the challenges faced by traditional single-objective optimization methods. By comprehensively considering various aspects of model performance, this research aims to enhance prediction accuracy, focusing on the model's generalizability and feasibility in practical applications. This approach provides a more comprehensive and reliable scientific basis for public health decision-making.

The main contributions of this study are as follows: First, we propose an infectious disease prediction model framework that incorporates multi-objective optimization. This approach achieves a balance among multiple performance indicators and enhances the overall predictive capability of the model. Secondly, through empirical research, we showcase the effectiveness of multi-objective optimization methods in enhancing the accuracy and stability of infectious disease predictions. Finally, the findings of this study offer new tools and insights for researchers in the field of infectious disease prediction and for public health decision-makers. This contributes to the scientific rigor and effectiveness of epidemic response strategies.

2/22

In summary, this study not only emphasizes the importance and application prospects of multi-objective optimization in infectious disease prediction but also highlights the urgency and significance of the research. Through this study, we aim to contribute to the development of infectious disease prediction models and provide stronger scientific support for global public health security.

## METHODOLOGY

### Data selection and processing

In this study, we utilized the Mexican COVID-19 time series dataset provided by Our World in Data (*Karlinsky & Kobak, 2021*). This dataset covers the period from April 1, 2020, to March 31, 2023, and includes various key indicators such as daily new confirmed cases (new_cases), total confirmed cases (total_cases), daily new deaths (new_deaths), total deaths (total_deaths), along with smoothed data and ratios calculated per million people. Such datasets are widely used in epidemiological research due to their completeness and accuracy. Based on the correlation analysis of the dataset, we selected indicators that are highly correlated with the daily new confirmed cases (new_cases) as the main variables (see Fig. 1). These indicators have significant predictive value in forecasting models, as identified by *Husnayain et al. (2021)*, *Mathieu et al. (2020)*, *Sharma et al. (2022)*, *Wang et al. (2022)*, *Zhang, Tang & Yu (2023)*.

To further analyze and enhance model performance, we performed a logarithmic transformation on the daily new confirmed cases (new_cases_log). Logarithmic transformation is a commonly used numerical processing technique that is effective in reducing the skewness of non-normally distributed data, as mentioned by *Vukašinović et al. (2023)*, *West (2022)*. We also conducted thorough data cleaning. For the data processing of each model, we followed the same steps. Initially, we used RobustScaler for data scaling, a normalization method known for its robustness to outliers, as noted by *Feng et al. (2014)*, *West (2022)*. Furthermore, we standardized the data to ensure the feature values conform to a distribution with a mean of 0 and a standard deviation of 1, as recommended by *Oka (2021)*, *Wang et al. (2023)*. In dividing the training and test sets, we used an 80% and 20% ratio, a practice supported by *Joseph (2022)*, who argue that this split effectively maintains the sequentiality and temporal coherence of time series data.

### Model selection

In this study, we selected several machine learning and deep learning models that are widely used in the field of time series prediction, based on their performance in existing literature and successful application in similar problems, as indicated by *Ahmed, Hassan & Mstafa (2022)*, *Lim & Zohren (2021)*.

### Feedforward neural networks

Feedforward neural networks (FNN) is a basic form of an artificial neural network, widely used in various machine learning tasks. FNN features include being simple and efficient, easy to implement, and debug. The FNN model structure and parameters are shown in Table 1.

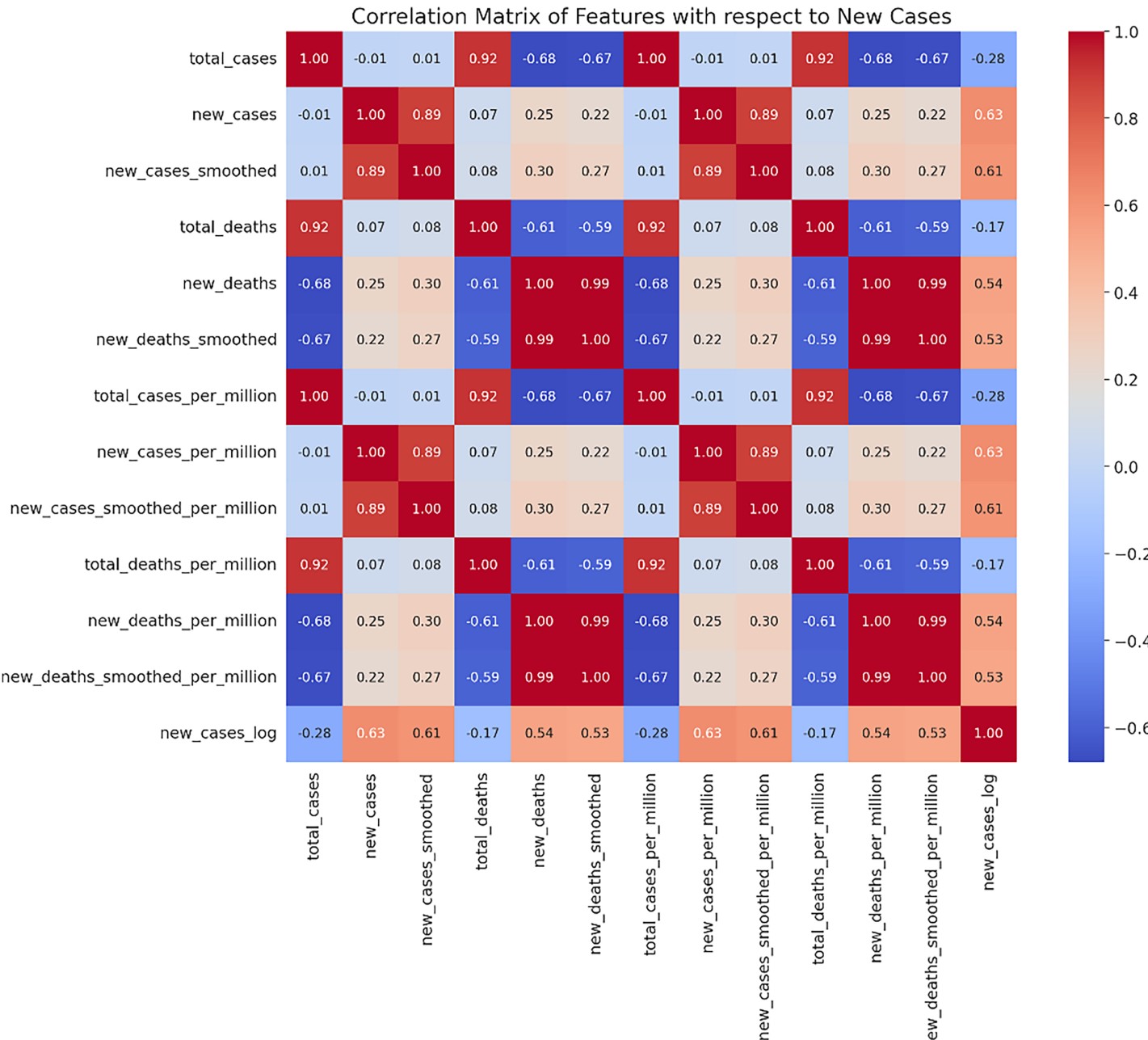

**Figure 1** **Correlation matrix heatmap of new cases.** A matrix where both rows and columns are labeled with the same set of features. The cells of the matrix are colored based on the correlation coefficient between two features. Red indicates a positive correlation, while blue represents a negative correlation.

## Convolutional neural networks

Convolutional neural network (CNN) is a neural network model widely used in the field of deep learning and is particularly effective at processing image data. It effectively recognizes patterns and features in images by mimicking the workings of the human visual system. Although originally designed for image analysis, CNNs have also been successfully applied

**Table 1 FNN model structure and parameters.**

| Parameter type | Descriptions |
|---|---|
| Model structure and parameter settings | Input layer: set according to the dimension of the feature data.<br>Fully connected layers: use dense layers with adjustable neuron numbers, employing the ReLU activation function.<br>Dropout layer: incorporate a dropout layer with a fixed ratio of 0.2 to reduce overfitting.<br>Output layer: use a single-neuron dense layer for predicting the target variable |
| Hyperparameter tuning | Learning rate options: 0.001, 0.01, 0.1<br>Neuron number options: 32, 64, 128<br>Batch size options: 16, 32, 64 |
| Search process | Iterate through various combinations, selecting the one that minimizes RMSE on the training set |
| Training configuration | Utilize the Adam optimizer, with a training duration of 100 epochs, setting the batch size according to the optimal parameters. |
| Optimal parameters | Learning rate of 0.001, 128 neurons, and a batch size of 16 |

to the processing of time series data. It shows potential for processing a wide variety of sequence data by learning local temporal features within the data. The CNN model structure and parameters are shown in Table 2.

## Long short-term memory networks (LSTM)

Long short-term memory (LSTM) is a deep learning model designed to address the long-term dependency issue in traditional recurrent neural networks (RNNs). LSTM effectively captures long-term relationships in time-series data by introducing special structural units, enabling it to demonstrate excellent performance in processing complex sequence data with temporal extensibility. The LSTM model structure and parameters are shown in Table 3.

## Temporal convolutional network

Temporal convolutional networks (TCN) offer a unique architecture well-suited for sequential input, especially in complex clinical decision support settings that involve time-series data. TCN gained popularity for its state-of-the-art performance across various applications. The TCN model structure and parameters are shown in Table 4.

## Random forest regressor

Random forest regressor (RF) is a widely used ensemble technique that utilizes a multitude of decision-tree classifiers. It operates on various sub-samples of a dataset with random subsets of features for node splits. This method enhances predictive accuracy and controls overfitting by using majority voting for classification problems or averaging for regression problems. Random Forest is particularly effective because of its ability to process large amounts of data with high accuracy. The RF model structure and parameters are shown in Table 5.

## Decision tree regressor

The decision tree regressor (DT) is a type of decision tree used for regression tasks. This model is renowned for its interpretability and effectiveness in capturing non-linear

**Table 2 CNN model structure and parameters.**

| Parameter type | Descriptions |
|---|---|
| Model architecture | Input layer: set input shape according to the feature dimensions of the time series data. |
| | Convolutional layer: use a single convolutional layer with adjustable numbers of filters and kernel sizes. |
| | Pooling layer: add a MaxPooling layer with a fixed pooling size of 2. |
| | Flatten layer: data outputted from the convolutional and pooling layers are transformed into one dimension through the Flatten layer. |
| | Fully connected layer: add a dense layer with adjustable unit numbers, employing the ReLU activation function. |
| | Output layer: a single-neuron Dense layer for predicting the target variable. |
| Training configuration | Optimizer: use the Adam optimizer. |
| | Learning rate: selected based on the results of a Keras Tuner search. |
| | Training epochs: train for 50 epochs on the training set. |
| | Batch size: set to 32 |
| Parameter search with Keras tuner | Define a model function using Keras Tuner's hyperparameters (hp) to define the model structure and search space |
| Configuration | Use randomSearch, targeting validation loss, with a maximum of 5 trials and 3 epochs per trial |
| | Search: conduct training for 10 epochs |
| Optimal parameters | conv_1_filter: 112, conv_1_kernel: 3, dense_1_units: 96, learning_rate: 0.001 |

**Table 3 LSTM model structure and parameters.**

| Parameter type | Descriptions |
|---|---|
| Model architecture | Input layer: set input shape according to the feature dimensions of the time series data. |
| | LSTM layer: utilize LSTM units with adjustable numbers. |
| | Dropout layer: added after the LSTM layer, with an adjustable ratio. |
| | Output layer: a single-neuron dense layer for prediction |
| Hyperparameter search | Search for combinations of different unit numbers, dropout ratios, and learning rates. Select the combination that minimizes RMSE on the test set.Parameter combination options: unit numbers (50, 100, 150), dropout ratios (0.2, 0.3, 0.4), learning rates (0.001, 0.0005, 0.0001). |
| Training configuration: optimizer | Use the Adam optimizer.learning rate: selected based on search results.Training epochs: train for 100 epochs on the training set.Batch Size: Set to 32. |
| Optimal parameters | Unit number: 100, dropout ratio: 0.3, learning rate: 0.001 |

**Table 4 TCN model structure and parameters.**

| Parameter type | Descriptions |
|---|---|
| Model architecture | Input layer: set according to the time series feature dimensions of the training data. |
| | TCN layer: utilize temporal convolutional network layers to process time series data, with parameters including the number of filters, kernel size, number of stacks, and dilation. |
| | Flatten layer: data outputted from the TCN layer is transformed into one dimension through the Flatten layer. |
| | Output layer: a dense layer using a linear activation function for predicting the target variable |
| Parameter setting and manual parameter search | Manually iterate through various parameter combinations, including different numbers of filters, kernel sizes, stack numbers, and dilation options.Select the combination that minimizes RMSE on the validation set. Parameter combination options: number of filters (32, 64, 128), kernel size (2, 3), number of stacks (1, 2), dilation options ((1, 2, 4, 8) and (1, 2, 4, 8, 16)) |
| Training configuration | Optimizer: use the Adam optimizer. |
| | Learning rate: Set to 0.002. |
| | Training epochs: train for 50 epochs on the training set. |
| | Batch size: Set to 16 |
| Optimal parameters | Number of filters: 64, Kernel size: 3, number of stacks: 1, dilation: (1, 2, 4, 8) |

**Table 5  RF model structure and parameters.**

| Parameter type | Descriptions |
|---|---|
| Hyperparameter search | Parameter grid: n_estimators (50, 100, 150), max_depth (None, 10, 20, 30), min_samples_split (2, 5, 10), min_samples_leaf (1, 2, 4).<br>Conduct parameter search using GridSearchCV, combined with 5-fold cross-validation, and the evaluation criterion being negative mean squared error |
| Optimal parameter | Max_depth None, min_samples_leaf 1, min_samples_split 2, n_estimators 50. |

**Table 6  Decision tree regressor model structure and parameters.**

| Parameter type | Descriptions |
|---|---|
| Hyperparameter search | Parameter grid: criterion ('squared_error', 'friedman_mse', 'absolute_error'), splitter ('best', 'random'), max_depth (None, 10, 20, 30, 40, 50), min_samples_split (2, 5, 10), min_samples_leaf (1, 2, 4).Conduct hyperparameter search using GridSearchCV, combined with 5-fold cross-validation |
| Optimal parameter | Criterion 'absolute_error', max_depth 10, min_samples_leaf 1, min_samples_split 2, splitter 'best'. |

**Table 7  XGBoost model structure and parameters.**

| Parameter type | Descriptions |
|---|---|
| Hyperparameter search | Parameter grid: n_estimators (50, 100, 200), max_depth (None, 10, 20, 30), learning_rate (0.01, 0.1, 0.2).Conduct hyperparameter search using randomizedsearchCV, combined with 5-fold cross-validation and 50 iterations |
| Optimal parameter | n_estimators 200, max_depth None, learning_rate 0.1. |

relationships in data. Decision tree regressors can handle both categorical and continuous input and output variables, making them versatile for a wide range of regression problems. The DT model structure and parameters are shown in Table 6.

### Extreme gradient boosting regressor (XGBoost)

Extreme gradient boosting (XGBoost) is a highly efficient and effective open-source implementation of the gradient boosting algorithm. It is particularly popular for its computational efficiency and strong performance in structured or tabular datasets for classification and regression predictive modeling problems. The XGBoost model structure and parameters are shown in Table 7.

### Ridge regression

Ridge regression, also known as Tikhonov regularization, is a method used to estimate the coefficients of multiple regression models in situations where linearly independent variables are highly correlated. It introduces a penalty term to the loss function: the squared magnitude of the coefficient multiplied by the regularization parameter. This approach is particularly useful in mitigating the problem of multicollinearity in linear regression models, thereby enhancing the model's prediction accuracy and interpretability. The ridge regression model structure and parameters are shown in Table 8.

**Table 8 Ridge model structure and parameters.**

| Parameter type | Descriptions |
|---|---|
| Hyperparameter search | Defaults |
| Optimal parameter | Defaults |

**Table 9  Model prediction performance metrics.**

| Name | Accuracy (RMSE) | Generalization (RMSE) | Computational efficiency |
|---|---|---|---|
| FNN | 152.025 | 4,532.84 | 26.19 s |
| TCN | 231.014 | 2,477.45 | 16.65 s |
| LSTM | 170.442 | 1,0307.7 | 35.16 s |
| CNN | 1,144.36 | 5,455.59 | 10.16 s |
| RF | 18.5805 | 1,251.51 | 0.43 s |
| DT | 24.4889 | 8.1421 | 0.07 s |
| xgboost | 33.0709 | 13.0672 | 0.38 s |
| Ridge | 11.5274 | 114.852 | 0.03 s |

## Multi-objective optimization

In this study, we focus on enhancing the performance of infectious disease prediction models through multi-objective optimization methods. Specifically, we aim to optimize the model's performance in three aspects simultaneously: the root mean square error (RMSE) of accuracy, the RMSE of generalizability, and computational efficiency (model training time). These three objectives are often conflicting; for example, enhancing accuracy and generalizability may reduce computational efficiency, leading to an increase in model training time (*Cui et al., 2022*; *Du et al., 2024*). Therefore, the aim of this study is to find the optimal trade-off among these three objectives. (see Table 9).

Performance metrics such as accuracy RMSE, generalizability RMSE, and computational efficiency (model training time) were acquired through the utilization of the chosen model, as delineated in Table 1. In this investigation, the NSGA-II (Non-dominated Sorting Genetic Algorithm II) within the genetic algorithm (GA) framework was opted for as the multi-objective optimization algorithm. The selection of NSGA-II was based on its efficacy in managing multi-objective optimization predicaments, particularly in upholding solution diversity and pinpointing the Pareto front (*Hu, Li & Liu, 2022*; *Liu, Ruan & Ma, 2023*; *Padilla-García et al., 2023*). Furthermore, the non-dominated sorting and crowding distance mechanisms of NSGA-II empower it to efficiently recognize a collection of optimal solutions in extensive search spaces. This capability is especially vital for determining the optimal trade-offs among diverse performance indicators in infectious disease prediction models (*Bolla et al., 2023*; *Entezari et al., 2023*; *Li et al., 2022*).

The model selection methodology employed in this research is executed through the NSGA-II algorithm, utilizing the DEAP (Distributed Evolutionary Algorithms in Python) library. The framework comprises the subsequent essential stages:

1) **Population initialization:** The population is initialized by randomly selecting algorithm parameters or model configurations.
2) **Fitness evaluation:** The evaluation function calculates the accuracy RMSE, generalizability RMSE, and computational efficiency (model training time) for each individual (*i.e.*, model configuration) in the population.
3) **Genetic operations:** Crossover (cxPassThrough) and mutation (mutate) operations are applied to generate new individuals, exploring the solution space.
4) **Selection mechanism:** The next generation of the population is selected using the selection mechanism (select) in the NSGA-II algorithm, based on non-dominated sorting and crowding distance.
5) **Iterative optimization:** Repeat the above process until a predetermined number of iterations or other stopping conditions are reached.

The experimental configuration consists of a population size of 100 individuals evolving over 50 generations, with a crossover probability of 0.7 and a mutation probability of 0.3. The optimization process is conducted through the utilization of the 'run_algorithm' function. The outcomes are then visually depicted using the 'plot_pareto_front_with_labels' function, which showcases the trade-offs between accuracy RMSE, generalizability RMSE, and computational efficiency (specifically model training time). This graphical representation serves to elucidate the interplay of various objectives and the efficacy of the NSGA-II algorithm in achieving a harmonious balance among them, thereby facilitating informed decision-making in the model selection process.

# RESULTS

## Multi-objective optimisation result

The outcomes of the multi-objective optimization conducted in this research can be effectively illustrated through a comprehensive examination using Pareto front analysis. As depicted in Fig. 2, the positioning of different models' performance in the multi-dimensional objective space is accurately represented, encompassing metrics such as the root mean square error (RMSE) of prediction accuracy, RMSE of generalizability, and computational efficiency (model training time). These parameters are used in graphical representations to assess and compare different prediction models, providing a clear visualization of how models handle the trade-offs among these conflicting objectives.

The Pareto front analysis visualizes the trade-offs between the following metrics:

1) **Accuracy RMSE:** This metric measures the precision of the model when predicting data. On the x-axis of the chart, a lower RMSE value indicates higher prediction accuracy.

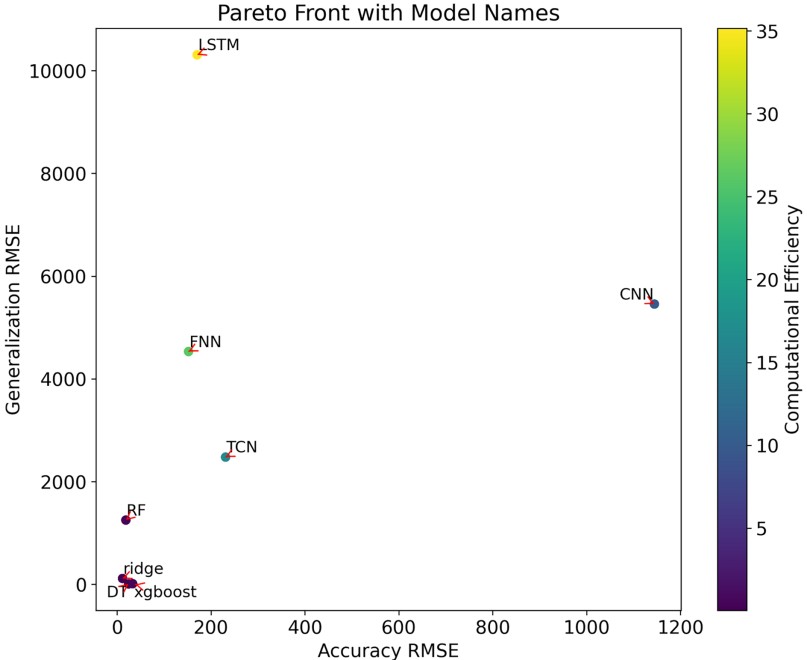

**Figure 2  Multi-objective optimisation of the Pareto frontiers.**

2) **Generalization RMSE:** This metric reflects the model's ability to generalize to new data. A lower RMSE value on the y-axis indicates better generalizability.

3) **Computational efficiency (model training time):** This is represented by a color bar, where the gradient (from purple to yellow) shows the change in computational efficiency from high to low.

In the chart, different points represent different models. The position of each point is determined by its RMSE values for accuracy and generalizability, while the color indicates its computational efficiency. Key observations include:

1) Models 'DT' and 'XGBoost' both exhibit lower RMSE values for accuracy and generalizability, while also maintaining relatively high computational efficiency (closer to purple), suggesting they may be well-balanced models optimizing accuracy, generalizability, and computational efficiency.

2) The 'Ridge' model shows lower values in terms of accuracy and computational efficiency but slightly lacks in generalizability compared to 'DT' and 'XGBoost'.

3) The 'LSTM' model shows the lowest generalizability and computational efficiency, while the 'CNN' model has the lowest accuracy.

These insights from the graphical representation help evaluate and compare the overall performance of different prediction models, assisting decision-makers in selecting the most suitable model based on specific needs and constraints.

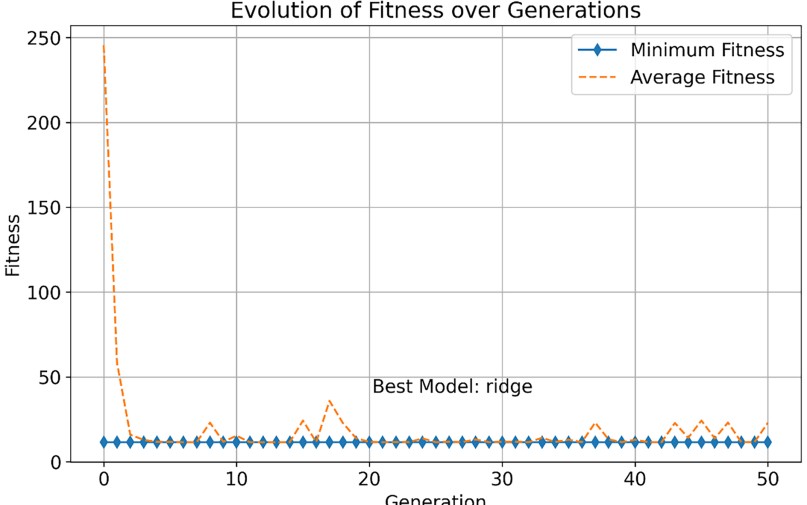

**Figure 3** **Evolution of fitness over generations.**

## Comparison of single-objective optimisation models

This study further explores the performance differences between multi-objective optimization methods and traditional single-objective optimization methods for selecting infectious disease prediction models. As shown in Fig. 3, we observed the change in fitness of the single-objective optimization method during the evolution process, which includes the evolution of minimum fitness and average fitness.

In the early stages, the minimum fitness decreases rapidly, indicating that the optimization algorithm can quickly identify and facilitate the evolution of high-performance solutions. This rapid evolutionary progress demonstrates the efficiency of the single-objective optimization approach in exploring the solution space. As generations increase, both the minimum and average fitness stabilize, demonstrating the robustness of the algorithm in consistently finding optimal solutions. The best model, "ridge," is identified after 50 generations, and it excels in all objectives, showcasing its superior overall performance.

## Case study

In order to validate the effectiveness of the method, we utilized the infectious disease dataset of COVID-19 in Indonesia and Iran to assess the performance of the model chosen through the multi-objective optimization method against the model selected through the single-objective optimization method in a real-world scenario. Through the multi-objective optimization approach, we selected the decision tree (DT) and gradient-boosted tree (XGBoost) as the best models. In contrast, the ridge regression model was chosen as the best model through the single-objective optimization approach.

In the actual scenario (refer to Table 10), the decision tree model demonstrates good performance with the lowest root mean square error (RMSE), indicating its high sensitivity to the data and strong predictive capability. Despite having a higher RMSE value compared to the decision tree model, the XGBoost model exhibits superior predictive performance.

**Table 10  Comparison of model performance in predicting RMSE for COVID-19 in Iran and Indonesia.**

| Model name | Iran (RMSE) | Indonesia (RMSE) |
| --- | --- | --- |
| DT | 8.1421 | 6.7367 |
| XGBoost | 13.0672 | 13.5128 |
| Ridge | 114.8524 | 16.2552 |

In contrast, the ridge regression model selected by the single-objective optimization method has a relatively high RMSE value on this dataset.

This case study demonstrates that a multi-objective optimization strategy may be better appropriate for picking a model with greater utility when dealing with a real-world challenge of predicting infectious illness data. The good performance of the decision tree model emphasizes the necessity of taking into account numerous performance measures when constructing a prediction model, rather than focusing exclusively on lowering prediction error.

# DISCUSSION

## The impact of model complexity on model performance

An increase in model complexity often improves the model's ability to fit. For example, by increasing the depth of the tree, a decision tree model can capture more features and patterns, thereby improving the prediction accuracy on the training set. However, models that are too complex tend to overfit, *i.e.*, perform poorly on test sets or new data because they may capture noise in the data rather than actual patterns (*Barea-Sepúlveda et al., 2023*). Similarly, XGBoost improves prediction accuracy by integrating multiple trees, *i.e.*, gradient boosting decision trees. Although an increase in complexity (*e.g.*, more trees, greater depth) often improves the accuracy of the model, it can also lead to overfitting (*Budholiya, Shrivastava & Sharma, 2022*).

Increasing the complexity of a model may reduce its generalization. For decision tree models, structures that are too complex perform poorly in the face of new data because they may have overfitted the training data. Moderate pruning and parameter tuning can help decision trees maintain good generalization on different datasets (*Kozyrev et al., 2023*). In the case of XGBoost, a modest increase in complexity can improve the generalization of the model, as XGBoost employs regularization techniques to prevent overfitting. However, overly complex models may still perform worse than training data on new data (*Hlongwane, Ramaboa & Mongwe, 2024*).

More complex decision trees require more computational resources to train and predict. A tree structure with a large depth will increase the computation time and storage requirements, which will affect the efficiency of the model (*Yang, Wang & Li, 2022*). Similarly, complex XGBoost models (more trees, more depth) can significantly increase computational time and resource requirements. Although XGBoost is optimized for computational efficiency, overly complex models still increase computational costs (*Silvestri et al., 2023*).

Increasing model complexity (*e.g.*, deeper trees, higher tree counts) often improves the accuracy of the training data, but this can lead to overfitting and reducing generality. Proper regularization and pruning techniques can help find a balance between accuracy and generalization (*Alalayah et al., 2023*). More complex models tend to have higher accuracy, but also require more computational resources and time. A trade-off between prediction accuracy and computational efficiency needs to be made based on the needs of the actual application scenario. For example, in applications that require real-time prediction, some accuracy may need to be sacrificed to ensure computational efficiency (*Papafotis, Nikitas & Sotiriadis, 2021*).

## Adaptability of different types of infectious diseases and different data characteristics

The multi-objective optimization approach has shown remarkable adaptability when processing data for different pathogen types. For example, in infections of different genotypes or serotypes, these methods can significantly improve the accuracy of inference for different pathogen types by optimizing laboratory surveillance networks. showed that by optimizing the HFMD surveillance network, the multi-objective optimization approach can significantly reduce the mean square error of estimating serotype-specific incidence, thereby improving the performance of the surveillance network (*Cheng et al., 2022*).

Different data features perform differently in the multi-objective optimization method. For example, when applying the multi-objective optimisation approach for COVID-19 prediction model selection, the generalisation capability of the model was incorporated, resulting in the selection of a model that not only performs well in COVID-19 prediction, but also performs well in terms of model performance in the prediction of new infectious diseases.

Some multi-objective optimization methods may not perform well when dealing with data-intensive, computationally complex problems. For example, while a multi-objective optimization approach can find a balance between different objectives, there may still be trade-offs between computational efficiency and data complexity. *Tan et al. (2017)* that although the multi-objective optimization method performs well in solving complex environmental/economic power dispatch problems, its computational complexity is still a major challenge.

## Comparison of different algorithms

When choosing a prediction model for infectious diseases, different multi-objective optimization algorithms have their own advantages and disadvantages;

1) **NSGA-II (Non-Dominant Sequencing Genetic Algorithm II)**; Good convergence and hybridization: NSGA-II is capable of generating high-quality Pareto leading-edge solutions for a wide range of optimization problems (*Liu, Ruan & Ma, 2023*). Excellent performance in many complex optimization problems, such as biological learning systems, traffic data analysis, *etc*. Shortcoming: In a multi-objective optimization problem, the selection pressure of NSGA-II decreases significantly as the number of

targets increases, affecting the convergence ability of the algorithm (*Kumari, Jain & Dhar, 2019*).

2) **SPEA2 (Intensity Pareto Evolution II):** SPEA2 excels at maintaining the diversity of solutions, maintaining the diversity of the Pareto front through intensity and distance measurements. It is especially suitable for design problems that require a high diversity of solutions (*Cai et al., 2022*). Shortcoming: Although SPEA2 performs well in terms of diversity, it may not converge as well as NSGA-II in some issues (*Babor et al., 2023*).

3) **MOEA/D (Decomposition-based Multi-Objective Evolutionary Algorithm):** MOEA/D optimizes multi-objective problems by decomposing them into multiple single-objective subproblems and solving them in parallel. This method excels in dealing with complex multi-objective problems, especially in high-dimensional object spaces (*Liu & Ye, 2023*). Shortcoming: MOEA/D may require more complex parameterization and higher computational resources for some problems (*Sun, 2023*).

## The long-term validity of the model and the impact of new data

Due to its simplicity and explanatory nature, decision tree models can effectively deal with complex data features in long-term forecasts (*Lange, 2023*). They can capture complex nonlinear relationships by recursively segmenting data, which can help with long-term prediction. Over time and as new data emerges, decision tree models may need to be updated frequently to maintain predictive performance (*Zhang et al., 2024*). These models perform mediocre in dealing with data drift because they are not flexible enough for emerging patterns. By integrating the advantages of multiple trees, XGBoost is able to capture more complex patterns and exhibit greater robustness to long-term data changes (*Li et al., 2024*). Although XGBoost has shown strong adaptability to new data, it still needs to be retrained regularly to ensure that the model adapts to changing infectious disease transmission patterns.

In the face of new data, the decision tree model may be difficult to adapt to the new model due to the fixed model structure. This requires frequent model updates and retraining to maintain prediction performance. Thanks to its gradient boosting mechanism, XGBoost is better able to adapt to new data. However, with the increase of data volume and the change of characteristics, it is still necessary to regularly adjust and optimize the hyperparameters to maintain high prediction accuracy.

## Challenges and strategies for integrating optimization models into public health decision-making systems

As new data continues to pour in, predictive models need to be updated and maintained frequently to ensure their accuracy and usefulness. This can be a challenge for public health agencies with limited resources. To do this, an automated data update and model retraining process is needed to ensure that the model can respond to new data and changes in a timely manner. At the same time, the necessary technical support and training are provided to improve the technical capacity of public health workers.

Effective deployment and use of predictive models requires cross-sectoral collaboration, including the involvement of multiple stakeholders, including governments, healthcare organizations, technology providers, and more. Lack of policy support and coordination mechanisms can lead to deployment failures. Therefore, it is necessary to establish a cross-sectoral collaboration mechanism and policy support framework to ensure that all parties can work closely together in the deployment and use of the model. At the same time, clear standards and guidelines should be developed to regulate the development and application of models.

## Advantages of multi-objective optimisation methods

This study demonstrates the significant advantages of the multi-objective optimisation approach in the prediction of infectious disease data. Firstly, the multi-objective optimisation approach significantly improves the usefulness and adaptability of the model by considering multiple evaluation criteria for model selection, rather than based on a single metric alone (*Khatun et al., 2022*). For example, the decision tree (DT) model and the XGBoost model not only perform well in terms of prediction accuracy, but also maintain high computational efficiency, showing excellent overall performance.

In addition, the multi-objective optimisation approach supports a comprehensive evaluation of the model, providing a framework for decision makers to weigh various performance metrics (*Zhao et al., 2024*). For high-risk decision support systems such as infectious disease prediction models, there is a real need to carefully balance the accuracy, generalisation ability and computational efficiency of the model in order to improve its usefulness, as seen in the case of COVID-19 outbreak data prediction in Indonesia.

A comparison of existing studies shows that although multi-objective optimization has been widely used and studied in other fields, relatively little research has been conducted on model selection for infectious disease prediction. Previous work has focused on improving a single performance metric, such as prediction accuracy, while insufficient attention has been paid to the generalization ability and computational efficiency of the model. In contrast, the methodology of this study not only considers prediction accuracy but also integrates other important performance metrics of the model, providing a more comprehensive evaluation framework for infectious disease prediction.

In this study, the multi-objective optimization methods employed surpass most existing methods in their ability to find the optimal equilibrium between multiple indicators. Furthermore, through empirical studies, we demonstrate that in real health crises, such as the COVID-19 pandemic, the utility of these methods far exceeds that of traditional single-objective optimization methods. This is particularly important in assessing the development of epidemics and guiding public health strategies.

## Practical implications of model selection

In the context of public health, the choice of infectious disease prediction models is related to the rational allocation of resources, the timely implementation of preventive measures, and the efficiency of emergency response (*Piscitelli & Miani, 2024*). Models selected by multi-objective optimisation methods, such as the DT and XGBoost models that perform

well in this study, provide policy makers with accurate and timely information on the development of epidemics due to the good balance between accuracy, generalisability and computational efficiency. This helps the government and public health organisations to develop more scientific and effective response strategies in the face of limited medical resources and the need to make quick decisions.

High-quality model predictions can enhance the accuracy of outbreak early warning systems, thereby guiding communities in the early stages of an outbreak to take measures to slow down the spread of the virus (*Cai et al., 2023*). For example, the ability of DT models to be understood and trusted by non-technical people due to their simplicity and easy-to-understand decision rules is a non-negligible advantage in outbreak management.

In real-world scenarios, models must also be able to adapt to changing data and sudden outbreak developments. In the case of this study, the DT model showed a high level of adaptability, which emphasises the importance of considering not only the accuracy of the predictions, but also the ability of the model to adapt to new data when selecting a model (*Zhang et al., 2023*). This flexibility in modelling is essential for monitoring outbreaks in real time and predicting their trends.

### Limitations of the study

In this study, a multi-objective optimisation approach was used in the development and evaluation of infectious disease prediction models, and although positive results were obtained, several limitations existed:

Firstly, only the COVID-19 dataset was used in this study, which limits the assessment of the generalisation ability of our models. Although the models selected by the multi-objective optimisation approach performed well on the COVID-19 dataset, we cannot confidently predict the performance of these models under other different infectious disease conditions.

Secondly, the NSGA-II algorithm was chosen as the tool for multi-objective optimisation in this study, which may have influenced the results of the optimisation. Other multi-objective optimisation algorithms may have produced different Pareto fronts and final set of models chosen, which implies that our findings were limited by the chosen algorithm. In addition, the experimental design and setup may also affect the interpretation of the results. For example, the choice of evaluation metrics may have an impact on the optimisation process and the final results.

### CONCLUSIONS

The primary goal of this research is to create and verify a multi-objective optimisation framework to improve predictive model selection for infectious disease data. By combining numerous criteria such as prediction accuracy, generalization ability, and computational efficiency, we verify DT and XGBoost as models that perform well on the COVID-19 dataset. These models outperformed models chosen using typical single-objective optimisation methods (*e.g.*, ridge regression) in terms of prediction accuracy, while also demonstrating a good balance of other performance indicators.

From a public health standpoint, our research emphasizes the significance of multi-objective optimisation methods in model selection for infectious illness prediction. As global health security faces new challenges, such as the COVID-19 pandemic, effective outbreak prediction models are crucial for developing public health strategies. Our findings give models that can help public health decision-makers better plan resource allocation, estimate the potential risk of epidemic waves, and implement appropriate preventative and control strategies.

Theoretically, this study shows that a multi-objective optimisation method works well when dealing with multi-dimensional performance indicators in predictive models. It offers a novel perspective that takes into account more than just one accuracy parameter in the model selection process, integrating accuracy, generalization capabilities, and computing efficiency into a holistic framework. This approach stretches the bounds of classic predictive model selection theory, resulting in a solution that is more appropriate for real-world challenges.

In practice, by taking into account multi-objective optimisation of forecasting models, this work delivers more refined and balanced forecasting tools for public health decision making. These technologies, especially during global health crises like the COVID-19 outbreak, can assist public health professionals in better predicting the development of outbreaks, developing more effective interventions, and optimizing resource allocation. The practical significance of this technique is not limited to current health concerns, but might be used to any field where several indicators must be merged for optimal decision-making.

Future research will be critical to enhancing the effectiveness and usability of predictive models for infectious illnesses. As new data emerges and prediction needs expand, new algorithms and approaches will be required to handle larger datasets and more diverse prediction jobs. Furthermore, investigating ways to more effectively incorporate expert knowledge and public health practice experience into a multi-objective optimisation framework would result in more accurate and useful prediction models.

### Funding
The authors received no funding for this work.

### Competing Interests
The authors declare that they have no competing interests.

### Author Contributions
- Deren Xu conceived and designed the experiments, performed the experiments, analyzed the data, authored or reviewed drafts of the article, and approved the final draft.
- Weng Howe Chan conceived and designed the experiments, performed the experiments, analyzed the data, authored or reviewed drafts of the article, and approved the final draft.

- Habibollah Haron analyzed the data, authored or reviewed drafts of the article, and approved the final draft.

## Data Availability

The data are available from Our World In Data (Mexico, Indonesia, Iran): https://ourworldindata.org/explorers/coronavirus-data-explorer?facet=none&country=~MEX&Metric=Confirmed+cases&Interval=New+per+day&Relative+to+Population=true&Color+by+test+positivity=true.

## Supplemental Information

Supplemental information for this article can be found online at http://dx.doi.org/10.7717/peerj-cs.2217#supplemental-information.

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
