# Peer review of "Enhancing infectious disease prediction model selection with multi-objective optimization: an empirical study"

_PeerJ Computer Science, doi:10.7717/peerj-cs.2217_

## Round 0.1 · original submission · Major Revisions

Collectively the reviewers have substantial concerns about this manuscript. The authors should provide point-to-point responses to address all the concerns and provide a revised manuscript with the revised parts being marked in different color.

·

Basic reporting

The study evaluates the effectiveness of applying multi-objective optimization methods to the selection of infectious disease prediction models by comparing them with models selected using traditional single-objective optimization methods. It particularly focuses on analyzing the performance of Decision Tree (DT) and XGBoost models in terms of accuracy, generalizability, and computational efficiency.

Experimental design

No comments

Validity of the findings

Here are my questions that could further explore the study’s findings:

1. How does the complexity of the Decision Tree and XGBoost models influence their accuracy, generalizability, and computational efficiency? Is there a trade-off between model complexity and performance across different metrics?
2. How adaptable are the multi-objective optimization methods when applied to different types of infectious diseases with varying data characteristics? Are there certain conditions under which these methods perform particularly well or poorly?
3. How might other multi-objective optimization algorithms, such as SPEA2 (Strength Pareto Evolutionary Algorithm 2) or MOEA/D (Multi-Objective Evolutionary Algorithm based on Decomposition), compare to NSGA-II in terms of effectiveness in selecting prediction models for infectious diseases?
4. How effective are the DT and XGBoost models selected via multi-objective optimization in predicting infectious diseases over a long term? Do these models maintain their performance as new data become available or as the disease evolves?
5. What are the challenges and considerations in integrating these optimized models into real-world public health decision-making systems? How can these models be deployed effectively to maximize their impact on public health policy and emergency response strategies?

Additional comments

Here are specific grammatical corrections and suggestions for the manuscript:

1. Abstract Section:
- Original: "This study aims to explore the application of multi-objective optimization methods in selecting infectious disease prediction models and evaluate their impact on improving prediction accuracy, generalizability, and computational efficiency."
- Suggested Correction: Add "the" before "application" for better flow: "This study aims to explore the application of multi-objective optimization methods in selecting infectious disease prediction models and to evaluate their impact on improving prediction accuracy, generalizability, and computational efficiency."
2. Methods Section:
- Original: "The NSGA-II algorithm was employed in this study to compare models selected using multi-objective optimization methods with those selected using traditional single-objective optimization methods through empirical research."
- Suggested Correction: Simplify and clarify the sentence: "In this study, the NSGA-II algorithm was used to compare models selected by multi-objective optimization with those selected by traditional single-objective optimization."
3. Results Section:
- Original: "The results indicate that decision tree (DT) and XGBoost models selected through multi-objective optimization methods outperform in terms of accuracy, generalizability, and computational efficiency."
- Suggested Correction: Correct the verb agreement and add "those" for clarity: "The results indicate that decision tree (DT) and XGBoost models selected through multi-objective optimization methods outperform those selected by other methods in terms of accuracy, generalizability, and computational efficiency."
4. Discussion Section:
- Original: "However, the limitations of this study also suggest future research directions, including improving algorithms, expanding evaluation metrics, and using more diverse datasets."
- Suggested Correction: Streamline and adjust tense for consistency: "However, this study's limitations suggest future research directions, including algorithm improvements, expanded evaluation metrics, and the use of more diverse datasets."
5. General Suggestions:
- Check for consistency in terminology (e.g., "multi-objective optimization" vs. "multiobjective optimization").
- Ensure all acronyms are defined at first use within the main text, even if they are common in the field.
- Use consistent terminology when referring to statistical terms or model names to avoid confusion.

Reviewer 2 ·

Basic reporting

This manuscript explores the feasibility of adopting the multi-objective optimization techniques to optimally select infectious disease prediction models. In the case study, authors selected DT (Decision Tree) and XGBoost models, based on the evaluation of the multi-objective optimization tool NSGA-II algorithm, in addition to ridge regression model based on the evaluation of the single-optimization method. The three models are evaluated on COVID-19 disease dataset. and results demonstrate outstanding performance for DT and XGBoost.


- The work is overall well prepared and written.
- I have some suggestions regarding the content:
o In Figure 2, please indicate the direction of highly desired value for each axis.
o In Case study Section, the term “RIDGE” was used without clearly indicating that it refers to Ridge regression model. I suggest to better replace this term with “ridge regression”.
o In the materials and methods Section, I suggest clearly stating that your work interest lies in regression problem (not classification)
o Affiliations 1 & 3 refer to the same information, I suggest omitting this duplicate.

Experimental design

no comments

Validity of the findings

no comments

Reviewer 3 ·

Basic reporting

no comment

Experimental design

no comment

Validity of the findings

Conclusions is not well stated or supported -
The authors claimed multi-objective optimization also “enhancing the performance of infectious disease prediction models”, this is no result supporting this. The authors should provide model performance comparisons with/without multi-objective optimization.

Additional comments

In general, this is a well written paper. The authors used multi-objective optimization method for model selection applied on Mexican COVID-19 time series dataset, and claimed DT and XGBoost performs best under metrics including accuracy, generalizability, and computational efficiency. With single objective method (accuracy RMSE), ridge model - with larger generalizability RMSE than DT and XGBoost - will be selected out instead.
However, I didn’t see the necessary of using multi-objective optimization for model selection in this application. First reason is, for small dataset, XGBoost (or maybe other DT based model) normally would be the best practice. Second, with mature and well packed machine learning packages, it would be very easy to try multiple models and compare multiple objectives - accuracy, generalization and computation efficiency (they are not taking time) for model evaluation. What's the significance of using multi-objective optimization?
Besides, in this paper the authors claimed multi-objective optimization also “enhancing the performance of infectious disease prediction models”, proving performance of all models comparing with/without multi-objective optimization would help to prove it. Maybe I have missed, but I didn’t see this result.

---

## Round 0.2 · accepted · Accept

The reviewers are satisfied with the revisions and suggest accepting this manuscript.

·

Basic reporting

The manuscript titled "Enhancing Infectious Disease Prediction Model Selection with Multi-Objective Optimization: An Empirical Study" examines the application of multi-objective optimization (MOO) methods in selecting predictive models for infectious diseases. The study employs the NSGA-II algorithm to compare models chosen by MOO against those selected through traditional single-objective optimization. Results indicate that models such as Decision Tree (DT) and Extreme Gradient Boosting Regressor (XGBoost) selected via MOO outperform single-objective models in terms of accuracy, generalizability, and computational efficiency. The study highlights the advantages of MOO in balancing multiple evaluation metrics, suggesting its potential to improve public health decision support systems.

Experimental design

No Comments

Validity of the findings

No more comments

Additional comments

No more comments